# CD24: A Marker for an Extended Expansion Potential of Urothelial Cancer Cell Organoids In Vitro?

**DOI:** 10.3390/ijms23105453

**Published:** 2022-05-13

**Authors:** Ruizhi Geng, Niklas Harland, Ivonne A. Montes-Mojarro, Falko Fend, Wilhelm K. Aicher, Arnulf Stenzl, Bastian Amend

**Affiliations:** 1Center for Medical Research, University Hospital, Eberhard Karls University, 72074 72072 Tuebingen, Germany; grz_880606@hotmail.com (R.G.); aicher@uni-tuebingen.de (W.K.A.); 2Department of Urology, University of Tuebingen Hospital, 72076 Tuebingen, Germany; niklas.harland@med.uni-tuebingen.de (N.H.); urologie@med.uni-tuebingen.de (A.S.); 3Institute for Pathology, Eberhard Karls University, 72074 Tuebingen, Germany; ivonne.montes@med.uni-tuebingen.de (I.A.M.-M.); falko.fend@med.uni-tuebingen.de (F.F.)

**Keywords:** bladder cancer organoids, immune checkpoint antigens, CD24, CD44, bladder cancer stem-cell marker

## Abstract

Background: Bladder cancer is the most cost-intensive cancer due to high recurrence rates and long follow-up times. Bladder cancer organoids were considered interesting tools for investigating better methods for the detection and treatment of this cancer. Methods: Organoids were generated from urothelial carcinoma tissue samples, then expanded and characterized; the expression of immune modulatory antigens and tumor stem cells markers CD24 and CD44 was explored in early (P ≤ 3) and later (P ≥ 5) passages (P) by immunofluorescence and by quantitative PCR of cDNA. The expression of these factors was investigated in the corresponding cancer tissue samples by immunohistochemistry. Results: The expression of the PD-L1 was detected on some but not all organoids. CD276 and CD47 were observed on organoids in all passages investigated. Organoids growing beyond passage 8 expressed both CD24 and CD44 at elevated levels in early and late cultures. Organoids proliferating to the eighth passage initially expressed both CD24 and CD44, but lost CD24 expression over time, while CD44 remained. Organoids growing only up to the 6th passage failed to express CD24 but expressed CD44. Conclusions: The data indicate that the expression of CD24 in urothelial cancer cell organoids may serve as an indicator for the prolonged proliferation potential of the cells.

## 1. Introduction

Bladder cancer (BC) is among the most common malignancies in many countries. A global incidence of more than 350,000 new cases and a prevalence of close to 3 million cases were estimated about a decade ago [1]. Current studies reported considerably higher incidences of more than 550,000 new cases per year [2]. The exposure of individuals to different pathogens and regular tobacco smoking have to be considered when discussing the cause of BC [3,4,5].

Besides behavioral and acquired risk factors, genetic disposition also contributes to BC [6]. Genome-wide screens including samples of cohorts of more than 3500 donors from several individual studies confirmed several gene loci associated with BC [7]. Mutations of genes common to most malignancies, for instance, factors controlling the cell cycle or cells death such as tumor protein P53, phosphatase and tensin homolog (PTEN), or loss of retinoblastoma (Rb) were also detected in BC samples [8].

Depending on the size and stage of development, tumors contain a blend of cells. A small cohort of cancer stem cells (CSC) is present in a specific tumor niche in most types of primary cancers [9,10]. These CSCs contribute to the ignition of tumor development and to the growth of the tumor itself [11]. However, true CSCs are mitotically quiescent and may remain for extended periods of time in a G0 stage of the cell cycle [12]. They therefore differ significantly from proliferating tumor cells in sensitivities towards anti-cancer drugs aiming at DNA replication or the regulation of the cell cycle and apoptosis [13,14]. Cells in G0 are also less sensitive to radiation and immunotherapy [15]. While radiation acts on cells with physical injury, chemotherapy and even more so immunotherapy provoke effects on active compounds on or in cells targeted on a biochemical level. Many biochemical processes can be modified by a corresponding blocking agent. This applies for inflammatory or immunological action against a tumor as well. The evasion of tumor cells from immunorecognition was associated with elevated expression of immune checkpoint molecules such as programmed cell death-1 (PD-1, CD279), its ligand PD-L1 (CD274), cytotoxic T-lymphocyte associated protein-4 (CTLA-4, CD152), or the cell surface molecule B7-H3 (CD276) [16]. These factors modulate the activation of T lymphocytes by the tumor and thus contribute to the immune tolerance against cancer cells. Another pathway modulates the intrinsic immune system by overexpression of the integrin-associated signal transducer CD47. Activated macrophages may induce apoptosis in tumor cells [17]. CD47 expressing tumor cells are not effectively attacked by macrophages. Blocking CD47 facilitated phagocytosis of cancer cells and growth of urothelial carcinoma cells in immune-deficient mice [18,19]. In addition, CD47 was also considered a bladder cancer stem-cell marker [14,20].

Cancer stem cells are recognized by the expression of distinct stem-cell markers. These markers differ in CSCs of tumors of different origins [11]. In solid tumors, the expression of the sialoglycoprotein CD24, the cell surface glycoprotein CD44, the transmembrane protein CD133, and the adhesion molecules CD166 (ALCAM) and CD326 (EPCAM) were associated with CSCs. However, CD133, CD166 and CD326 were considered CSC markers for colorectal cancer [21,22]. The expression of CD133 in BC samples was discussed, but its expression was not confined to urothelial BC cells in vitro [23,24]. CD166 mutants were associated with an increased BC susceptibility, but expression of CD166 on BC stem cells was not explored [25]. As mentioned above, CD47 is considered a BC stem-cell marker [20], and CD24 as well as CD44 were considered CSC markers for BC as well [20,26]. In animal tumor models, interactions of CD24 with P-selectin facilitated contact of tumor cells with the vasculature [27], and CD24 seems to play an important role for metastases [28]. Another study investigated the expression of CD133 and CD24 on BC cells and concluded that a CD133^pos^CD24^pos^ phenotype was associated with tumor progression and metastases [29]. However, CD24 as a BC stem-cell marker was questioned recently, indicating that this topic requires additional research [30]. CD44 expression was associated with homing of leukemic CSCs to their stem-cell niche [31]. Detection of CD44^pos^ cells may therefore be associated with CSC and stem cell niches in corresponding samples. At the same time, organ-specific CSC markers eventually open new avenues for diagnosis and therapy. In addition, analyses of the expression of these markers may facilitate the identification of CSCs in in vitro cultures as well.

In this study, we therefore investigated the expression patterns of immune checkpoint molecules PD-L1, CD276, and the immune modulatory molecule CD47 on bladder cancer organoids (BCO) in early versus late passages in comparison to the bladder cancer stem-cell markers CD24 and CD44 on protein and transcript levels. The corresponding tumor samples were characterized as well.

## 2. Results

### 2.1. Expression of Immune Checkpoint Antigens on Early and Late Passage 3D Organoids

Organoids BCO#140, #154 and #147 were established from urothelial carcinoma cells and passaged in vitro. Some key clinical data of the patients are listed in Table 1. BCO#140 yielded stable cultures expandable for more than 28 passages, which corresponded to 10 months of continued growth in vitro. BCO#154 generated cultures expandable for 20 passages, then growth decelerated. Cells of BCO#147 failed to proliferate for more than eight passages and ceased further cell growth. The expression of immune modulatory antigens and cancer stem-cell markers was explored in early (passage 0–3) and later stages (passage 5–11). In a first set of experiments, we stained organoids in chamber slides in their three-dimensional (3D) structure to investigate the gene expression patterns of the cells in the original positions in organoids (Figure 1, Figure 2, Figure 3 and Figure 7). In a second set of analyses, paraffin sections were generated from additional samples of the same organoids to compare the gene expression of individual cells in more detail (Figure 4, Figure 5, Figure 6 and Figure 8).

In early passage organoids in chamber slides, CD276 was detected in more than 70% of the cells, while PD-L1 was not detected or expressed only on a few cells. The majority of cells expressed CD47 (Figure 1, Figure 2 and Figure 3; Table 2). In late passage organoids, comparable expression patterns were observed (Figure 1, Figure 2 and Figure 3; Table 2). Differences in the expression patterns of immune modulatory antigens among the stable and expandable BCO#140 in comparison to the less-stable BCOs were not observed. Detailed antigen staining patterns of representative organoids are presented online (Appendix A).

To explore the expression of the immune checkpoint antigens and immune modulatory CD47 on an individual cellular level and in more detail, paraffine sections were generated. CD276 and CD47 were found to be prominently expressed in the cells of BCO#140, #147, and #154 (Figure 4, Figure 5 and Figure 6, Table 3). The expression patterns of the immune checkpoint molecule CD276 and the “do-not-eat-me signal” CD47 presented remarkable patterns. In BCO#140, CD276^+^ and CD47^+^ cells were distributed randomly in the organoids (Figure 4 and Figure 5). Only a few double-positive cells were noted in early and late passage cells. BCO#140 and BCO#154 were clusters filled with cells. By contrast, urothelial carcinoma cells of patient #147 generated mainly hollow speres (Figure 6). At least in the early-stage organoids, a distinct expression pattern of CD47 and CD276 was noted. On the same cell, CD276 was found on the outside of the organoid, while CD47 was found in the inside. This polarity became visible as in early stages the BCO#147 spheres grew in large part as monolayers (Figure 6). This pattern was maintained to some degree through later stages on spheres.

### 2.2. Expression of Bladder Cancer Stem-Cell Markers CD24 and CD44 on Early and Late Passage 3D Organoids

The expression of bladder cancer stem-cell markers CD24 and CD44 was investigated on organoids as well. While the expression of CD44 was observed in all three organoids, in early and later stages of in vitro culture, the expression of CD24 was remarkably different (Figure 7, Table 4). BCO#140 presented a prominent signal of CD24 in early-stage cultures, and even in later passages, some cells expressed CD24 (Figure 7). Only a few CD24^+^/CD44^+^ double-positive (DP) cells, visualized by yellow fluorescence, were found. Overall, the red fluorescent CD44^+^ cells were enriched in an outer ring while the green fluorescent CD24^+^ cells were found in an inner ring, separated partly by a layer of CD24^−^/CD44^−^ double-negative (DN) cells (Figure 7).

In contrast, in BCO#154, only a few cells expressed CD24 in early-stage cultures. As seen in BCO#140, CD24^+^ cells were not found frequently in the outer zone of the organoid but enriched in the center (Figure 7). On BCO#154, CD24^+^/CD44^+^ DP urothelial carcinoma cells were not detected (Figure 7). Moreover, in later-stage BCO#154, expression of CD24 ceased, but the organoids maintained a prominent expression of CD44 (Figure 7, Table 2). The cells in BCO#147 failed to express CD24 even in early passages but expressed CD44 in early- as well as in later-stage cultures (Figure 4, Table 3).

The analysis of expression of CD24 and CD44 in paraffin sections of the same organoids yielded somewhat different results (Figure 8). As described above, prominent expression of CD24 was observed on BCO#140 and BCO#154 in early-stage and later-stage cultures. In BCO#147 samples, CD24 was not detected. CD44 was detected in all samples (Figure 8, Table 4). Overall, this corroborated that BCO#147 was not enriched for cells expressing stem-cell-marker CD24, and this correlated with moderate expansion capacities of organoids from this donor, while BCO#140 and BCO#154 expressed CD24, and this correlated with an extended expansion capacity (Table 4 and Table 5). Of note, BCO#147 generated hollow spheres, which was not observed in BCO#140 and BCO#154 (Figure 8).

### 2.3. Transcripts Encoding the Immune Modulatory Factors and Stem-Cell Markers

In addition, we investigated the expression of the immune modulatory cell surface proteins and stem-cell markers on the transcript levels on BCOs. Early-passage (HL19/16) and late-passage (Hblac) normal human urothelal cells served as controls (Figure 9). Transcripts encoding PD-L1, CD276, and CD47 were detected in the normal urothelial cells and in BCOs in all samples investigated, and only minor differences were computed (Figure 9). The expression of BC stem-cell markers CD24 and CD44 was investigated on transcript levels as well (Figure 9). A reduced expression of CD24 was noted in late-stage normal urothelial cells when compared to early-stage urothelial cells (Figure 9). A comparable trend was observed in BCO#140 and BCO#154 organoids in early versus late cultures. Most importantly, the highest expression of CD24 transcripts was determined in early-stage BCO#140 (Figure 9). Of note, BCO#147 expressed CD24 transcripts in the later passage, although protein expression was not detected on cultured cells or parafin sections of organoids (Figure 7 and Figure 8). In general, transcripts encoding CD44 were slightly below CD24. In late-passage normal cells, the trend of reduced expression of CD44 was observed, while in early versus late BCO#140 and BCO#154, no differences were found (Figure 9).

RNA was extracted from organoids of BCO#140, #154, and #147 in early (plain bars) and late (dashed bars) passages as indicated to generate cDNA for quantitative PCR. Human urothelial cells HL19/16 and Hblac served as normal early- and late-passage controls, respectively. The y-axis presents the transcript amounts normalized to two house-keeping genes; the x-axis shows the target gene investigated as indicated.

### 2.4. Expression Marker Proteins on Tissue Samples of the Corresponding Bladder Cancer

The expression of the immune modulatory antigens and the stem-cell markers were investigated in the corresponding ex vivo tumor samples of patients #140, #154, and #147 as well (Figure 10). The expression of PD-1 and PD-L1 was observed by immunohistochemistry only on a few cells of BC tissue samples, but a prominent signal was observed for CTLA-4 (Figure 10). In all paraffin sections investigated, some cells expressed CD276, but only weak signals were observed for CD47 expression in one of three bladder-cancer samples (Figure 10). The expression of stem-cell markers CD24 and CD44 staining was seen only in a few individual cells, but not in considerable areas of the samples (Figure 10).

To correlate the expression patterns observed by organoids in vitro with the pathological situation in bladder cancer tissue, paraffin sections were stained with antibodies to PD-1, PD-L1, CTLA-4, CD276, CD47, CD24, and CD44 as indicated, and detected by immunohistochemistry. Tissues were counter-stained by HE. Size bars indicate 100 μm.

## 3. Discussion

In this study, we investigated the expression of immune modulatory antigens and CSC markers CD24 and CD44 on BCOs in early and late passages. More than 30 years ago, CD24, then termed NALM-6M1, was described as a cell-surface molecule on pre-B-lymphocytes, and transcripts were found to be enriched in tumor cells which carried functional mutations of tumor suppressor gene p53 [32,33]. CD24 is a small adhesion molecule (80 amino acids, 8082 Dalton molecular mass). It was shown to promote the invasion of tumor cells [34]. CD24 is highly expressed on bladder cancer samples, and it was associated with tumor cell proliferation, the potential of colony-forming units in soft agar cultures, and the regulation of apoptosis [35,36]. Analysis of clinical data provided evidence that the expression of CD24 correlated significantly with a rather short remission-free survival of the patients and an increased risk of metastases [28,35]. These data are in line with our results, as elevated expression of CD24 in early- and later-stage organoids was evident in the long proliferating BCO#140 cells, but not detectable in line BCO#147 on protein levels in both early- as well as late-passage organoids. In addition, the highest transcript levels encoding CD24 were enumerated in early-stage BCO#140 cells, correlating to its relevance for organoid stability in vitro. The mechanisms involved in higher BCO stability may include the effects of CD24 on cell–matrix interactions [37]. In cell-attachment assays, CD24 facilitated cell binding to fibronectin, to collagens I and IV, and to laminins through α3βb1 and α4β1 integrins [38]. Matrigel—employed to generate the organoids in this study—is prepared from murine sarcoma supernatants and contains about 60% laminins, 30% collagen IV, and some other components. CD24^high^ urothelial carcinoma cells may therefore attach better to this scaffold when compared to CD24^low^ cells [38]. Integrin binding to the extracellular matrix may modulate pro-apoptotic signals [39]. Integrin signaling may even reduce the sensitivity of cancer cells to apoptosis-inducing chemicals [40,41]. However, in pancreatic cancer cells, low expression of CD24 correlated with augmented adhesion to Matrigel, while elevated CD24 correlated with reduced adhesion [42]. This suggests that the attachment of CD24^high^ bladder cancer cells to the extracellular matrix in organoids may depend not only on β1 integrin-mediated binding but possibly also on additional factors.

Insufficient cell binding of adherently growing cells such as epithelial cells in spheroids is associated with apoptosis [43]. This may explain the instability of organoids by increased loss of CD24^high^ cells in the organoids. Moreover, CD24 was described as a mucin-like adhesion molecule, specifically binding to P-selectin, but not to E- or L-selectins [37]. Therefore, CD24 may contribute to direct cell-to-cell binding in organoids. However, the expression of P-selectin (CD62P) was not observed in normal bladder tissue [44], but elevated soluble P-selectin was found in the serum of bladder cancer patients [45]. As serum from patients is not complementing our cell culture media, P-selectin seems not to be critical for actions of CD24 on BCO stability and growth discussed here. Cross-linking CD24 induces apoptosis in a variety of cells [46,47]. However, this requires a corresponding cross-linking ligand and therefore probably does not play a role in our experimental setup. Taken together, we conclude that CD24^high^ bladder cancer cells are protected in organoids from CD24-induced apoptosis as urothelial cells express the CD24-ligand P-selectin at low levels, while at the same time elevated soluble P-selectin in the serum of bladder cancer patients prevents CD24 cross-linking on the tumor cells, and at the same time apoptosis of CD24^high^ cells in tumor tissues.

We noted that the CD24 transcript levels in late-stage normal urothelial cells Hblac were clearly lower when compared to less expanded normal urothelial cells HL19/16. The same trend was observed in BCO#140 and BCO#154 cells. Although transcripts encoding CD24 were detected in all late stage BCOs investigated, our results suggest that expression of the CD24 depends on culture time or number of passages of BCOs in vitro. This finding is corroborated by studies employing serial cell expansions to investigate the number of CD24^pos^ cells [48]. Moreover, experimental knock-down of CD24 expression in human bladder cancer cell lines reduced sphere formation, sensitivity to cisplatin-induced apoptosis, reduced the expression of the tumor stem-cell marker CD133, and attenuated tumor growth in an in vivo cancer model [26]. An invasive phenotype was associated in BC with CD133^pos^CD24^pos^ co-expression [29]. In contrast, in mesenchymal breast cancer-derived cell lines the invasive phenotype was associated with a CD44^pos^CD24^neg^ phenotype [49]. This conflict of data may be resolved by the hypothesis that CD24^high^ cells in our organoids are epithelial/urothelial cells but not mesenchymal cells based on their expression of cytokeratins [50]. In addition, expression of CD24 on urothelial carcinoma is androgen regulated [51]. A loss of CD24 expression on BCOs over time of in vitro expansion may be associated with a reduced level of androgen signaling through androgen receptors [51]. As both BCO#140 and BCO#147 were derived from male donors, major differences in androgen signaling are not expected. Detailed experiments along these lines on larger cohorts including BC samples from male and female donors must be addressed in future studies. Taken together, elevated expression of CD24 may serve as indicator for promising expansion of bladder cancer organoids.

The role of CD44 in the context of BC stem-cell markers seems less clear [20,52] and an androgen regulation was suggested [53]. We confirm the expression of CD44 on BCOs on transcript and protein levels in early and later passages and in situ in BC tissue samples. Recent animal studies corroborated a low expression of CD44 on 3D bladder cancer organoids, which was upregulated by propagation of the cells in so-called 2.5D organoid constructs in later passages [14,54,55]. We assume that the role of CD44 in BCOs requires further investigations.

Other results also merit discussion. By immunohistochemistry, we detected expression of PD-1 and CTLA-4 on paraffin sections of tumor samples. PD-1 and CTLA-4 are considered immune checkpoint molecules on T lymphocytes [56]. Proliferation of T cells requires IL-2. However, this cytokine is not added to our BCO expansion medium. Accordingly, by RT-qPCR, PD-1 transcripts were not detected at all or recorded at the detection threshold (data not shown). In contrast, CTLA-4 staining was prominent in all tumor samples investigated. This indicated that CTLA-4^pos^ cells were enriched in bladder cancer tissue. In the BCOs, CTLA-4 transcripts were barely above the detection threshold of this technique and two logs below the transcript amounts recorded for PD-L1 (data not shown). This result was unexpected as CTLA-4 is a well-known T lymphocyte antigen. However, recently it was shown that CTLA-4 can be expressed on thymic epithelial cells upon functional loss of the autoimmune regulator AIRE [57]. We therefore cannot exclude that some BCO cells transcribed mRNA encoding this T-lymphocyte associated gene. However, this remains to be investigated in future experiments.

Due to the sensitive technology employed for immunofluorescence, the signal intensities shown in this study should not be interpreted as quantitative data in the sense of describing expression levels or protein amounts in or on individual cell. Comparably, the staining patterns presented in the tables should also not be interpreted as indicators for staining intensities in or on individual cells in organoids, but strictly taken as indicator of the percentage of cells showing in staining. This challenge in data interpretation becomes evident when staining patterns of whole organoids in 3D structures are compared to the patterns obtained from the same organoids and passage by paraffin sections. One has to take into account that staining of whole organoids in 3D may detect primarily proteins on cell surfaces, while on paraffin sections, antibodies bind to both intracellular antigens and membrane-bound targets. Moreover, in organoids, fluorescence from the whole construct may add up to a collective staining. In contrast, in thin paraffin sections, less antigen mass is accessible. These differences may contribute to the different signal frequencies observed between 3D organoid staining in comparison to staining of paraffin sections. However, investigating BCOs by fluorescence microscopy in their 3D structure as well as in paraffine sections provided another advantage as well. The hollow structure of BCO#147 was clearly demonstrated in paraffine sections, but it was not evident from immunofluorescence analyses of organoids in 3D. Some difference between BCO#140 and BCO#154 and BCO#147 were also observed for expression of BC stem-cell marker CD24 in the three BCOs investigated here.

While most BCO studies published to date investigated the expression of cytokeratins, proliferation, and tumor markers in vitro in comparison to the tumor in vivo [58,59,60], information of the dynamics of expression of immune checkpoint antigens on BCOs is more limited. However, such data may become a valuable tool to screen for patient-specific checkpoint therapies when the regulation of these cell surface proteins is explored in more detail. At present, we continue to investigate these differences between staining of whole organoids and the corresponding paraffin sections with the transcript levels. In addition, we complement these studies by immunohistochemistry of cryosections.

One of the limitations of this study is the low number of individual BCOs investigated at the time of exploration. We intended to study the expression of immune checkpoint antigens and bladder cancer stem-cell markers in BCOs in early as well as late passages. Sufficient material was available from many tumor samples. Cells from many BC tissue samples were seeded in Matrigel. However, after initial growth, some samples failed to generate three-dimensional constructs resembling bona fide BCOs but rather grew in cellular clusters with fuzzy rim and without convincing mitotic activity. In addition, several BCOs retarded proliferation in passages 3 to 5. Only a few passed this crisis, including BCO#140, BCO#147, and BCO#154. Other BCOs passed beyond passage 5 but did not provide sufficient material for molecular analyses at later stages. To overcome these limitations in the future, we investigated all possibilities to improve the efficacies of BCO production and long-term culture. Moreover, only surgical samples from male tumor patients were included. In future studies, tumor samples from female patients and from patients representing clinical pTa to pT4 BC tumors should be included.

Bladder cancer-derived organoids have been and will be part of intensive research in tumor biology [58,59,61,62]. They reflect the tumor situation much better when compared to standard 2D cultures and may even help to better comply with refinement, reduction, and even in part replacement of studies employing living animals in pre-clinical tumor research. We recently initiated experiments to use organoids to investigate in vitro some of the physical and chemical characteristics of BC and to facilitate the discrimination between normal and malignant bladder tissue [63].

## 4. Materials and Methods

### 4.1. Bladder Cancer Organoids

Pathologist confirmed urothelial carcinoma samples were obtained from patients undergoing surgery after written and informed consent. The study was approved by the local Ethics Committee under file number 804/2020/B02. Tumor cells were isolated from the tissue by mechanical degradation followed by proteolytic digest and seeded in hydrogel domes as described recently [59]. In brief, the tissue was minced and collected by centrifugation (480 g, 10 min. ambient temperature (a.t.)). The sediment was resuspended in buffer containing collagenase (3000 U/mL) and hyaluronidase (1000 U/mL), and incubated under moderate agitation (37 °C, 30 min). The proteolytic degradation was continued by adding fresh collagenase for 30 min at 37 °C. Debris was removed by a cell strainer (70 μm mesh) and the filtrate was sedimented by centrifugation (150 g, 7 min, a.t.). The yield and viability of the cells were enumerated, resuspended at 4E06/mL to obtain 4E04 cells in 10 μL. These cells were mixed on ice with 30 μL Matrigel, dipped in a 24-well plate and flipped upside-down to generate hanging drops. After a short incubation at 37 °C, the plates were turned back, complemented with 500 μL BTM culture medium per well, and incubated in a cell-culture incubator (37 °C, 5% CO_2_, humidified atmosphere). All organoid cultures included in this study were downscaled to match 48-well plates. Normal human urothelial cells HL19/16 in early passage and commercial urothelial cells in late passage (Hblac, CELLnTEC) served as controls [64]. A detailed protocol is disclosed in Appendix B.

### 4.2. Immunofluorescence and Immunohistochemistry

To characterize the BC organoids, two distinct methods of immunofluorescence were applied. For three-dimensional evaluation of the BCOs, organoids were cultured in 8-well chamber slides. Immediately prior to staining, the medium was aspirated and the BCOs were rinsed twice with PBS. BCOs were fixed by 4% formaldehyde (30 min, a.t.), washed well three times with PBS, blocked (5% BSA, 0.2% Triton X-100, 0.1% Tween 20, in PBS; 1 h, a.t.), and incubated (1 h, 37 °C, humidified chamber, dark) with primary antibodies to immune checkpoint antigens PD-L1, CD276, to immunoregulatory molecule CD47, as well as to bladder cancer stem-cell markers CD24 and CD44. Unbound primary antibodies were washed away (3 × 5 min, PBS, a.t.). Primary antibodies were detected by incubation of the samples with complementary fluorescence-labelled secondary antibodies (1 h, a.t., humidified chamber, dark). Unbound secondary antibodies were rinsed away (3 × 5 min., PBS, a.t.). Cellular nuclei were counterstained by DAPI, and the expression of the marker genes was visualized by microscopy (Leica Stellaris 8 or Zeiss Axiophot). Antibody diluent was 1% BSA in PBS. Samples omitting the primary antibodies and samples stained with mouse or rabbit IgG isotype antibodies served as controls (data not shown). Details on the antibodies employed are listed in Appendix C.

For two-dimensional analyses of BCO paraffin, sections were generated following standard procedures. BCOs were expanded in 48-well plates to reach the culture density desired. The medium was aspirated and BCOs were rinsed twice by PBS and fixed (4% formaldehyde, overnight (o.n.), a.t.). The BCOs were dehydrated and embedded in paraffin. Paraffin sections were generated (3 μm, Leica RM2125, a.t.), dewaxed and rehydrated by aid of incubation in xylene, rehydrated by ethanol in decreasing concentrations, and mounted on glass slides as described [65]. The samples were incubated with the primary and secondary antibodies as described above. Cellular nuclei were counterstained by DAPI, and the expression of the marker genes was visualized by microscopy (Zeiss Axiophot). Samples omitting the primary antibodies and samples stained with mouse or rabbit IgG isotype antibodies served as controls (data not shown). Details on the antibodies employed are listed in Appendix C.

Immunohistochemistry of paraffin sections was utilized to detect expression of the immune modulatory antigens and bladder cancer stem-cell markers in bladder cancer tissue samples from the patients corresponding to #140, #154, and #147. In addition to the organoid samples, tumor samples were stained with antibodies to PD-1 and CTLA-4 (Appendix C). The primary antibodies were washed off (3 × for 5 min, PBS), and detected by the HRP or AP polymer reagent (IHC ZytochemPlus) The samples were counterstained by HE, covered (VectaMount, Vectorlabs), and recorded by microscopy (Axiovert A1, Zeiss).

### 4.3. Transcript Analyses

Quantification of gene expression on mRNA transcript levels was performed as described [66]. In brief, the cells were washed and counted. Nucleic acids were extracted, DNA was degraded enzymatically, and total RNA was isolated using a kit (RNeasy, Qiagen, Hilden, Germany). The yield and purity of total RNA were determined by UV spectroscopy (Nanodrop; Implen, Munch, Germany). A measure of 1μg of total RNA was reverse transcribed employing oligo-(dT) primers (Advantage RT-for-PCR kit TakaraBio, Saint-Germain-en-Laye, France). Quantitative PCR (LightCycler 480; Roche, Pensberg, Germany) of cDNA corresponding to the target genes was performed as described recently [67]. The primer sequences are listed in Appendix D. Quantification of transcripts encoding GAPDH and β-actin served as controls, and amounts of target transcripts were normalized in each batch to the two controls as described [67].

## 5. Conclusions

We conclude that elevated expression of CD24 on bladder cancer organoids may serve as an indicator for the in vitro culture and expansion capacities of BCOs. This notion is supported by our results on transcript and protein levels. Studies to investigate the diagnostic and therapeutic potential of CD47 and CD276 by aid of BCOs seem feasible, while the expression patterns observed for CTLA-4 on BCOs require additional research. Using three distinct BCOs is the limitation of our experiments. The study will therefore be complemented by additional research.

## Figures and Tables

**Figure 1 ijms-23-05453-f001:**
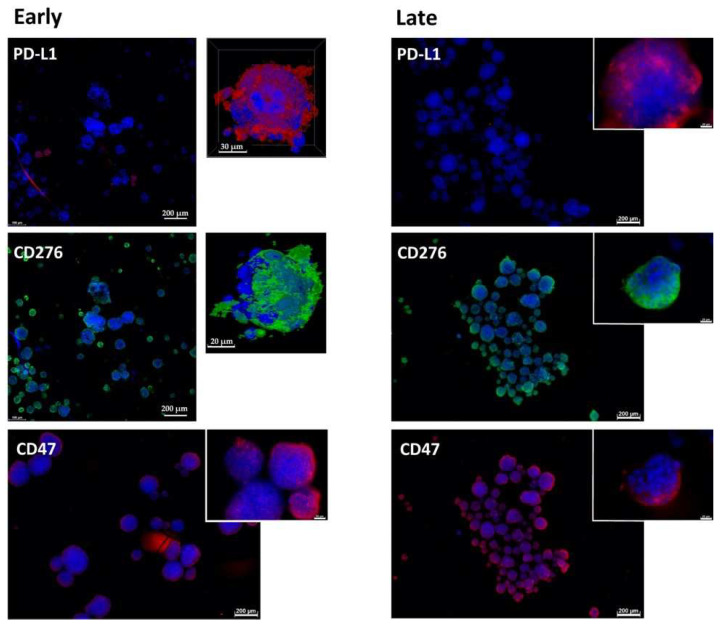
Expression of immune checkpoint antigens on BCO#140 organoids. Organoids of BCO#140, a population with high in vitro growth capacity, were fixed and directly stained in early passage (**left panel**) or in later passage (**right panel**) cultures with antibodies reactive with PD-L1, CD276, and CD47 as indicated. For each staining, an overview with several organoids and a detailed picture of individual organoids are presented. Size bars indicate 200 μm and 20 μm, respectively.

**Figure 2 ijms-23-05453-f002:**
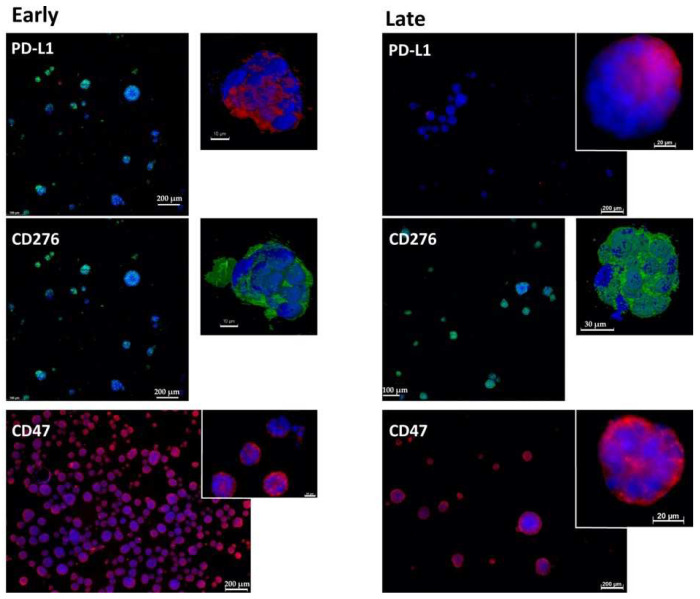
Expression of immune checkpoint antigens on BCO#154 organoids. Organoids of BCO#154, a population with moderate in vitro growth capacity, were fixed and directly stained in early passage (**left panel**) or in later passage (**right panel**) cultures with antibodies reactive with PD-L1, CD276, and CD47 as indicated. For each staining, an overview with several organoids and a detailed picture of individual organoids are presented. Size bars indicate 200 μm, 100 μm, 20 μm, and 10 μm as indicated.

**Figure 3 ijms-23-05453-f003:**
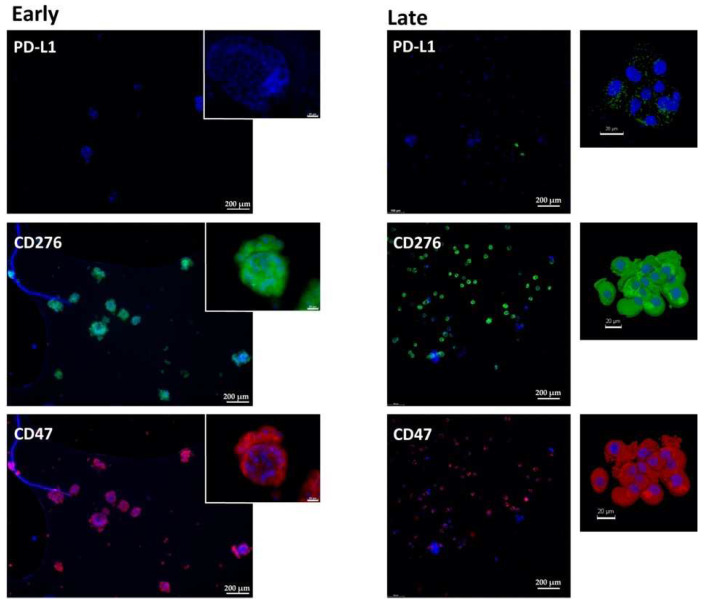
Expression of immune checkpoint antigens on BCO#147 organoids. Organoids of BCO#147, a population with limited in vitro growth capacity, were fixed and directly stained in early passage (**left panel**) or in later passage (**right panel**) cultures with antibodies reactive with PD-L1, CD276, and CD47 as indicated. For each staining, an overview with several organoids and a detailed picture of individual organoids are presented. Size bars indicate 200 μm and 20 μm, respectively.

**Figure 4 ijms-23-05453-f004:**
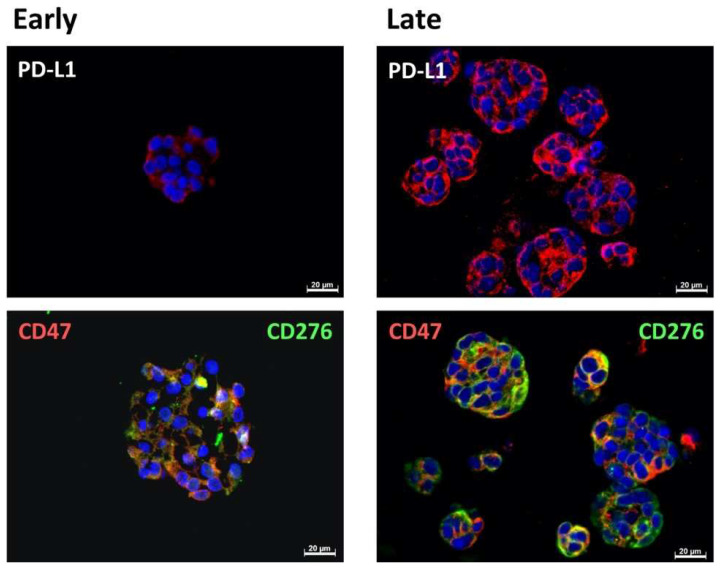
Expression of immune checkpoint antigens on BCO#140. Paraffin sections of organoids of BCO#140, a population with high in vitro growth capacity, were generated and stained in early passage (**left panel**) or in later passage (**right panel**) with antibodies reactive with PD-L1, CD276, and CD47 as indicated. In the bottom panel, CD47 is visualized by red fluorescence, CD276 by green fluorescence. Size bars indicate 20 μm, respectively.

**Figure 5 ijms-23-05453-f005:**
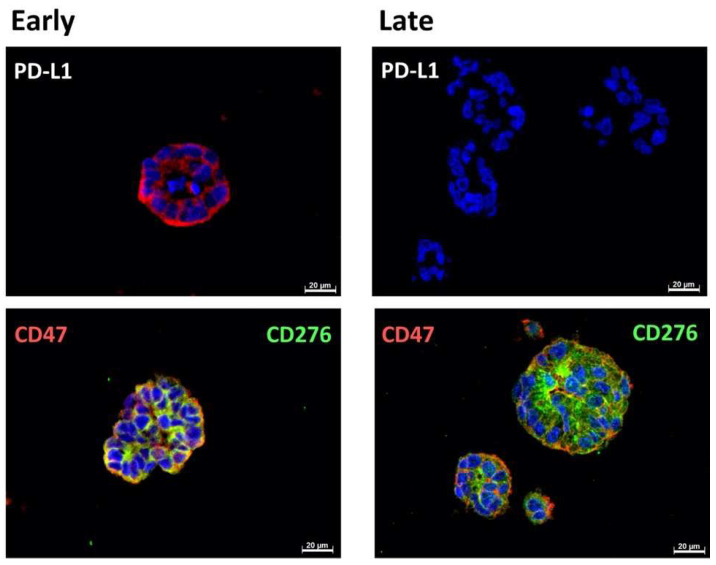
Expression of immune checkpoint antigens on BCO#154. Paraffin sections of organoids of BCO#154, a population with moderate in vitro growth capacity, were generated and stained in early passage (**left panel**) or in later passage (**right panel**) with antibodies reactive with PD-L1, CD276, and CD47 as indicated. In the bottom panel, CD47 is visualized by red fluorescence, CD276 by green fluorescence. Size bars indicate 20 μm, respectively.

**Figure 6 ijms-23-05453-f006:**
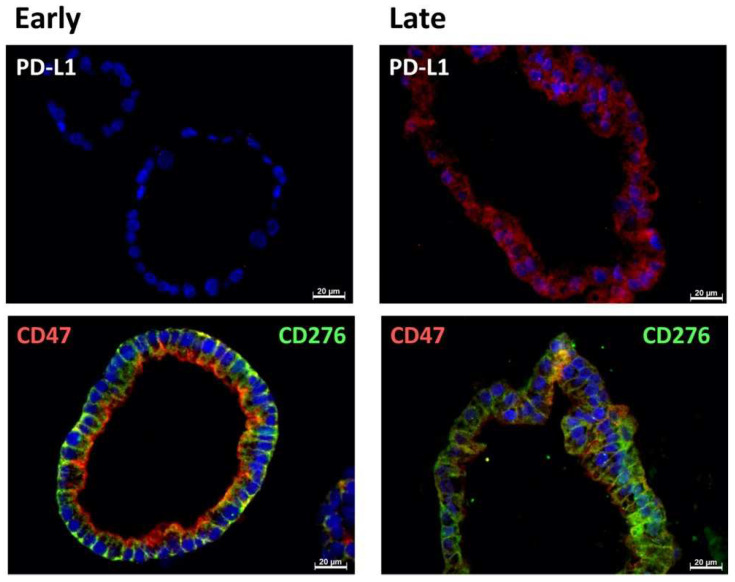
Expression of immune checkpoint antigens on BCO#147. Paraffin sections of organoids of BCO#147, a population with limited in vitro growth capacity, were generated and stained in early passage (**left panel**) or in later passage (**right panel**) with antibodies reactive with PD-L1, CD276, and CD47 as indicated. In the bottom panel, CD47 is visualized by red fluorescence, CD276 by green fluorescence. Size bars indicate 20 μm, respectively.

**Figure 7 ijms-23-05453-f007:**
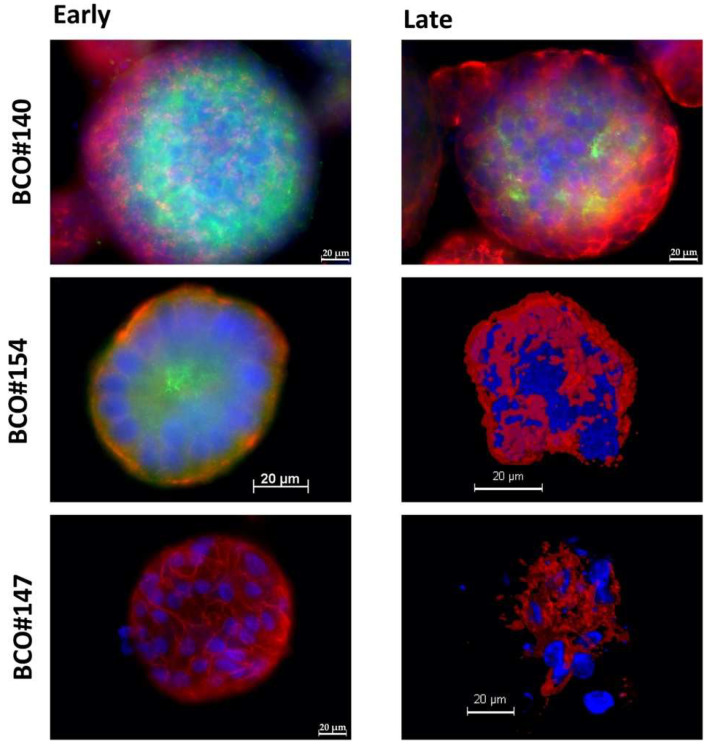
Expression of CD24 and CD44 on organoids. Organoids of BCO#140, #154, and #147 were fixed and stained with primary antibodies reactive with bladder cancer stem-cell markers CD24 and CD44. Expression of CD24 was visualized by a green fluorescent AlexaFluor488-labelled detection antibody. CD44 was visualized by a red fluorescent Cy3-labelled detection antibody. Cell nuclei were stained by DAPI.

**Figure 8 ijms-23-05453-f008:**
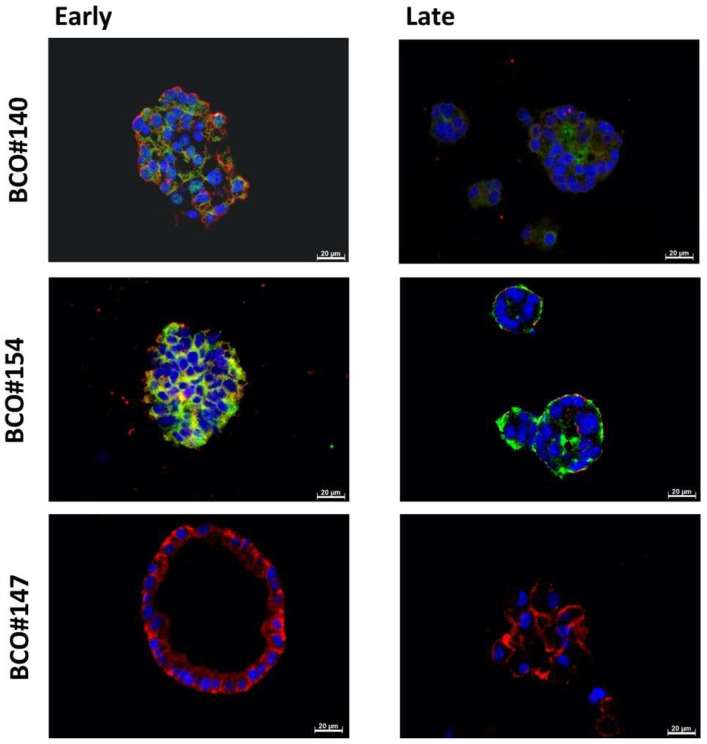
Detection of CD24 and CD44 on paraffin section of organoids. Paraffin sections of BCO#140, #154, and #147 were generated and stained with primary antibodies reactive with bladder cancer stem-cell markers CD24 and CD44. Expression of CD24 was visualized by a green fluorescent AlexaFluor488-labelled detection antibody. CD44 was visualized by a red fluorescent Cy3-labelled detection antibody. Cell nuclei were stained by DAPI.

**Figure 9 ijms-23-05453-f009:**
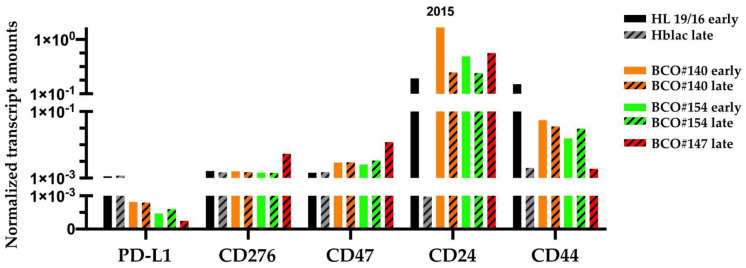
Enumeration of transcripts encoding immune modulatory antigens and stem-cell markers.

**Figure 10 ijms-23-05453-f010:**
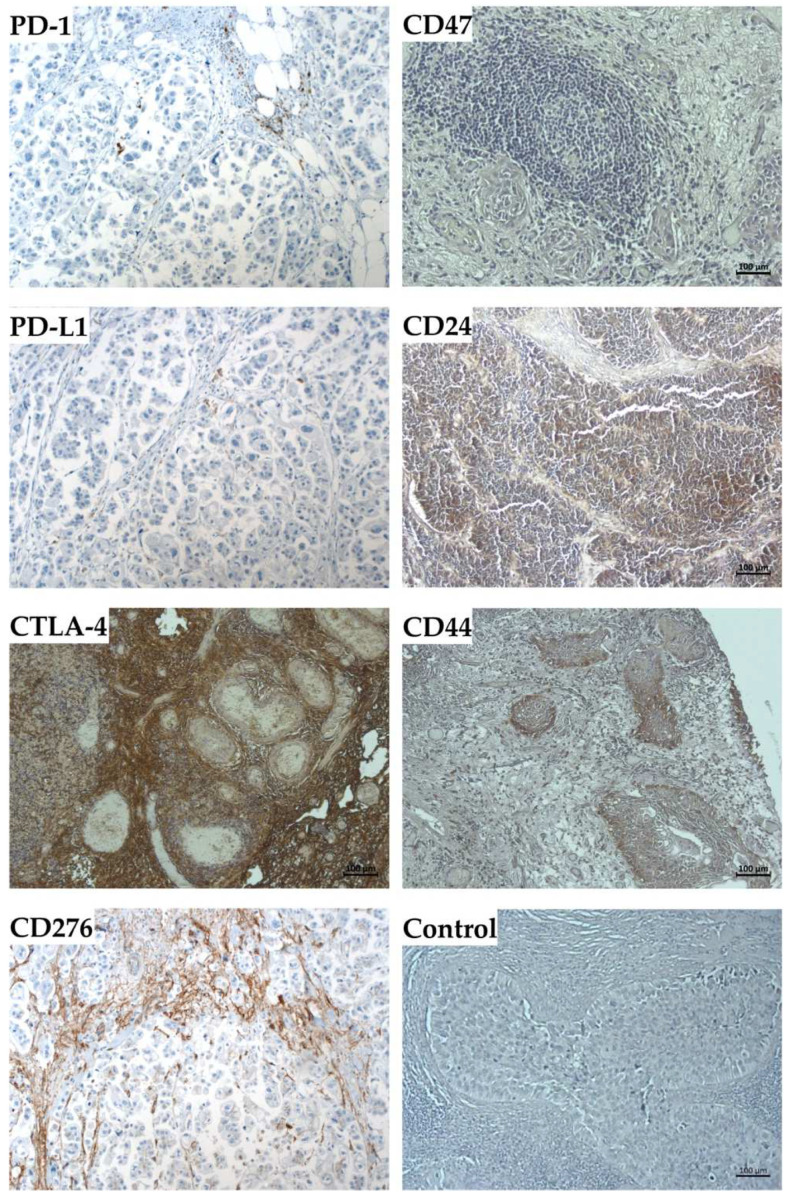
Detection of immune checkpoint antigens and stem-cell markers in bladder cancer tissue.

**Table 1 ijms-23-05453-t001:** Information on tumor samples included.

Patient	#140	#147	#154
Age (years)	64	61	64
Gender	male	male	male
Localization	bladder	renal pelvis	bladder
Tumor Stage (pT) Tumor Grade (G)	pT1 G3, high grade	pT4 G3 high grade	pT2 G3 high grade
Surgery	TURBT ^1^	RNU ^1^	RC ^1^
Histology	Small cell carcinoma	Urothelial carcinoma	Urothelial carcinoma

^1^ TURBT = transurethral resection of a bladder tumor, RNU = radical nephroureterectomy, RC = radical cystectomy.

**Table 2 ijms-23-05453-t002:** Expression of immune regulatory antigens on bladder cancer organoids in chamber slides ^2^.

Cell	Passage	PD-L1	CD 276	CD 47
BCO#140	Early	+	3+	3+
	Late	+/−	3+	3+
BCO#154	Early	+/−	3+	3+
	Late	+/−	3+	3+
BCO#147	Early	−	3+	3+
	Late	+/−	3+	3+

^2^ The expression ratio of antigens in early (P ≤ 3) and late (P ≥ 5) passage organoids was counted by fluorescence microscopy. The percentage of positive cells were scored as follows: +/−: < 5%, +: 5~30%, 2+: 30~70%, 3+: >70% of total cells. Total numbers of cells were visualized by DAPI and counted accordingly.

**Table 3 ijms-23-05453-t003:** Expression of immune regulatory antigens in paraffin sections of bladder cancer organoids ^3^.

Cell	Passage	PD-L1	CD 276	CD 47
BCO#140	Early	pos	pos	pos
	Late	pos	pos	pos
BCO#154	Early	pos	pos	pos
	Late	neg	pos	pos
BCO#147	Early	neg	pos	pos
	Late	pos	pos	pos

^3^ The expression ratio of antigens from early (P ≤ 3) and late (P ≥ 5) passages of organoids. Due to the method applied, the expressions of factors investigated are described as positive (pos) versus negative (neg) but not quantified by any means.

**Table 4 ijms-23-05453-t004:** Expression of stem-cell markers on organoids of BC in chamber slides ^4^.

Cell	Passage	CD 24	CD 44
BCO#140	Early	2+	3+
	Late	2+	3+
BCO#154	Early	+	3+
	Late	−	3+
BCO#147	Early	−	3+
	Late	−	3+

^4^ The expression ratio of antigens in early (P ≤ 3) and late (P ≥ 5) passage organoids was counted by fluorescence microscopy. The percentage of positive cells was scored as follows: +/−: < 5%, +: 5~30%, 2+: 30~70%, 3+: >70% of total cells. Total numbers of cells were visualized by DAPI and counted accordingly.

**Table 5 ijms-23-05453-t005:** Expression of stem-cell markers in paraffin sections of BC organoids ^5^.

Cell	Passage	CD 24	CD 44
BCO#140	Early	pos	pos
	Late	pos	pos
BCO#154	Early	pos	pos
	Late	pos	pos
BCO#147	Early	neg	pos
	Late	neg	pos

^5^ The expression ratio of antigens from early (P ≤ 3) and late (P ≥ 5) passages of organoids. Due to the method applied, the expressions of factors investigated are described a positive (pos) versus negative (neg) but not quantified by any means.

## Data Availability

Data will be provided to colleagues in public research institutions upon specific request.

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
