# Peer review of "CD24: A Marker for an Extended Expansion Potential of Urothelial Cancer Cell Organoids In Vitro?"

_ijms, 2022, doi:10.3390/ijms23105453_

Round 1
Reviewer 1 Report
General comments
The authors of this manuscript explored the expression of several markers such as CD24, CD44, CD47, CD276, and PD-L1 in urothelial cancer cell organoids at early and late passages. They showed that CD24 may serve as an indicator for a prolonged proliferation potential of the cells. The manuscript to me is, in general, clearly written. The science and technical execution of the study is of good quality. The study is solid and the data, in general, support the conclusions. The theory, logic, and design are easy to follow and in general make sense. However, some comments are necessary to improve the quality of the manuscript.
Specific comments
- In section 4.1: what do you mean by 4E06/mL, 2E04 cells?
- More details about sample information are necessary, like whether the tumors were muscle-invasive or not, the ages and gender of the patients, …etc.
- Bladder cancer is very common, why the authors analyzed only 3 strains.
- The study is focusing on marker expression in urothelial cancer. It should be supported with recent references:
- Page 2, at the end of the 2nd paragraph besides reference 20, add this more recent reference: https://www.mdpi.com/2073-4409/9/1/235/htm
- At Page 2, at the end of the 3rd paragraph line 10, (and CD24 as well as CD44 are well-described CSC markers for BC20.) support this sentence with more recent references such as: https://doi.org/10.1080/15384047.2021.1919004 ; https://www.nature.com/articles/s41598-020-66229-w ; https://onlinelibrary.wiley.com/doi/full/10.1111/cas.14118
- 7 title: CD24 and CD24 !!. it is CD44.
- Table 1: hoe the quantification was done?
- Fig legends: indicate the scale bars, it is not clear on the images.
- Add one figure showing the bright field images of the organoids at early and late passages.
- The word degrade for the matrigel is not suitable.
- Always show the centrifugation speed in g speed, not rpm.
Overall, I believe the improved version of the manuscript will be of interest to the field of PDOs for precision medicine and cancer biology. Therefore, it should be recommended for publication in IJMS after revision.
Author Response
General comments
The authors of this manuscript explored the expression of several markers such as CD24, CD44, CD47, CD276, and PD-L1 in urothelial cancer cell organoids at early and late passages. They showed that CD24 may serve as an indicator for a prolonged proliferation potential of the cells. The manuscript to me is, in general, clearly written. The science and technical execution of the study is of good quality. The study is solid and the data, in general, support the conclusions. The theory, logic, and design are easy to follow and in general make sense. However, some comments are necessary to improve the quality of the manuscript.
Specific comments
- In section 4.1: what do you mean by 4E06/mL, 2E04 cells?
Ad 1: We thank for pointing to this typographical error. The word “2E02” was wrong in the original version. We corrected it during revision.
- More details about sample information are necessary, like whether the tumors were muscle-invasive or not, the ages and gender of the patients, …etc.
Ad 2: We agree to this suggestion and added this information in the new table 1 in the revised manuscript. We corrected the numbering of the other tables in the manuscript R1 accordingly.
- Bladder cancer is very common, why the authors analyzed only 3 strains.
Ad 3: This point is well taken, and we addressed this question in a new paragraph at the end of the discussion (lines 477 – 490 in the PDF “changes”) The challenge is, that only a few organoids met our inclusion criteria, which are 1) sufficient material in early passage bona fide BCOs available for analyses of gene expression on transcript levels as well as analyses by immune fluorescence of organoids in 3D and by paraffin sections. 2) sufficient material available from the same organoids after later passages.
- The study is focusing on marker expression in urothelial cancer. It should be supported with recent references: Page 2, at the end of the 2ndparagraph besides reference 20, add this more recent reference: https://www.mdpi.com/2073-4409/9/1/235/htm At Page 2, at the end of the 3rd paragraph line 10, (and CD24 as well as CD44 are well-described CSC markers for BC20.) support this sentence with more recent references as: https://doi.org/10.1080/15384047. 2021.1919004 https://www.nature.com/articles/s41598-020-66229w ; https://onlinelibrary.wiley.com/doi/full/10.1111/cas.14118
Ad 4: We are happy to add the publications suggested to the revised manuscript, which all present excellent and topical experimental work. However, we prefer to discuss questions concerning BC stem cell markers not in the introduction and therefore added this information and the citations suggested to the discussion of the revised manuscript (compare lines 429 – 434 in the PDF “changes”).
- The title: CD24 and CD24 !!. it is CD44.
Ad 5: We prefer to focus in the title on CD24, as the expression of this protein was detected at distinct levels on the 3 BCOs investigated. Moreover, its expression correlated to the expansion potential of the organoids. But we added the term CD44 to the list of key words, as CD44 is an interesting cell surface antigen on BCOs as well.
- Table 1: how the quantification was done?
Ad 6: Some information for quantification was missing in the legend of table 1 in the original manuscript. We therefore revised the legend of table 2 (was previously table 1) in version R1 to better explain it. The legend of table 4 in the revised manuscript R1 was corrected accordingly.
- Fig legends: indicate the scale bars, it is not clear on the images.
Ad 7: We do agree to this comment and reply as follows: Scale bars were generated by the program of the microscopes automatically. They are visible on the original high-resolution micrographs. However, to generate a usable PDF for the review process we generated a low-quality PDF with only 72 dpi resolution, which generated already a 3.7MB document. However, to include this important information in the manuscript upon acceptance, we improved the visibility of the size bars in the revised set of figures and submit an improved PDF.
- Add one figure showing the bright field images of the organoids at early and late passages.
Ad 8: As the bright field image of an organoid does not give any information on changes in gene expression patterns, we prefer not to add additional artwork this manuscript. Additional figures require additional paragraphs at least in Results, Discussion and M&M. We are afraid to distract from the focus of this study.
- The word degrade for the matrigel is not suitable.
Ad 9: We think that this reviewer points to line 770 of Appendix B on the original version. We thank the reviewer for addressing this detail. We replaced the word “degrade” by “digest” as trypsin is the active component at this experimental step.
- Always show the centrifugation speed in g speed, not rpm.
Ad 10: When referring to centrifugations, the gravitational force (g) was disclosed in the original version of the manuscript where applicable. In the original document the term “rpm” was used only once (see line 905 in the PDF “changes). The term “rpm” was used there only to describe the rotation of a shaker, but not the g force of a centrifuge. We therefore prefer to stick with “rpm” in this context.
Overall, I believe the improved version of the manuscript will be of interest to the field of PDOs for precision medicine and cancer biology. Therefore, it should be recommended for publication in IJMS after revision.
Reviewer 2 Report
The authors described a very interesting and important organoid model for bladder cancer and investigated stem cell as well as immune markers on these spheroids. Such models represent important tools to further understand tumor development as well as possible therapeutic options. The data, especially by immunofluorescence, are very impressive. There are some points to improve:
- Detailed data on tumor samples have to be provided: are these muscle-invasive tumors? TNM and grading should be given for each sample.
- Primary tumors consist on different cell types including not only tumor cells but also fibroblasts. The cell content of spheroids should be investigated and described. Are these really only epithelial cells?
- The authors should describe the purpose of such models: what are the further plans to use these models?
- Introduction should be more focused on the issue of the study. It is not necessary to describe general knowledge on predisposition. Furthermore, the part on stem cell markers should be improved, especially the language stile. In addition, it is not clear why the authors aimed to investigate PD-L1 and CD276 in correlation to the two stem cell markers. This has to be clarified.
Author Response
The authors described a very interesting and important organoid model for bladder cancer and investigated stem cell as well as immune markers on these spheroids. Such models represent important tools to further understand tumor development as well as possible therapeutic options. The data, especially by immunofluorescence, are very impressive. There are some points to improve:
- Detailed data on tumor samples have to be provided: are these muscle-invasive tumors? TNM and grading should be given for each sample.
Ad 1: We thank this reviewer for this very important suggestion. A new table 1 was added presenting the information requested.
- Primary tumors consist on different cell types including not only tumor cells but also fibroblasts. The cell content of spheroids should be investigated and described. Are these really only epithelial cells?
Ad 2: We totally agree to this point and several studies have disclosed the composition of organoids by e.g., cytokeratin-positive urothelial/epithelial cells, vimentin-positive mesenchymal cells, and others. However, to avoid a lack of focus (the revised version of the manuscript comes with 10 figures, consisting of 4 – 8 elements each and 5 tables in the main document. In addition, 2 tables and detailed protocols are found in the appendix), we prefer to explore specifically the immune-checkpoint antigens and bladder cancer stem cell markers on protein and transcript levels at early and late passage BCOs. But our current and future studies investigate of course the phenotypically diversity of BCOs in more detail.
- The authors should describe the purpose of such models: what are the further plans to use these models?
Ad 3: Organoids may become helpful in vitro models to investigate the tumor biology. They are currently used in experiments for anti-cancer drug screening and development of novel tumor diagnosis tools. We appreciate this comment and added a short section along these thoughts to the revised manuscript (see lines 492 – 497 in PDF “changes).
- Introduction should be more focused on the issue of the study. It is not necessary to describe general knowledge on predisposition. Furthermore, the part on stem cell markers should be improved, especially the language stile. In addition, it is not clear why the authors aimed to investigate PD-L1 and CD276 in correlation to the two stem cell markers. This has to be clarified.
Ad 4: The Special Issue to which this manuscript was submitted will present paper representing different fields of bladder cancer research. A literature search by the key words (quote) “immune checkpoint antigens; CD24; CD44; … “ will find our manuscript – upon acceptance and publication. We therefore prefer to give the colleagues working in quite distinct areas such as biology of CD24 or CD44 a short introduction to the background of our study. Still, a moderate revision was performed, and the new introduction is overall approximately 10% shorter when compared to the primary text (see side bars page 1 and 2 in the PDF “changes”).
Reviewer 3 Report
The authors aimed to address the role of CD24 in urothelial carcinoma organoids. They conclude that the expression of CD24 in urothelial cancer cell organoids may serve as an indicator for a prolonged proliferation potential of the cells.
Several questions still need to be addressed:
- Previous publications showed that CD24 is not suitable for the direct detection of cancer stem cells in urothelial carcinoma. However, CD24 is proved to be significantly related to urothelial carcinoma grade and stage, which are both important prognostic indicators. The novelty of this study is limited.
- Variation of this biomarker is also noted in organoids. There are only three samples in the study which is not enough.
Author Response
- Previous publications showed that CD24 is not suitable for the direct detection of cancer stem cells in urothelial carcinoma. However, CD24 is proved to be significantly related to urothelial carcinoma grade and stage, which are both important prognostic indicators. The novelty of this study is limited.
Ad 1: We thank this reviewer for this comment. As stated in the title, we present CD24 primarily as marker for the expansion potential of BCOs. Our experiment were therefore not designed to explore CD24 as stem cell marker or to study the pathological aspects of CD24 in the bladder of patients in situ on in vivo. Still, we included representative experiments from ex vivo studies as well (see Fig. 10). But the data in 9/10 figures and in 4/5 tables document an in vitro situation only. In addition to CD24 (and CD44), we explored the expression patterns of immune checkpoint antigens on BCOs as function of in vitro passages. Such data have to the best of our knowledge not been published yet.
The role of CD24 as marker for expansion capacities of BCOs in vitro seems not to be contradictory to the results to which this reviewer points: CD24 and carcinoma grade and stage. However, several studies discussed CD24 in the context of BC stem cells (see e.g., Hofner et a. 2014, Ooki, 2018). This must be appreciated in such a study. But a recent paper challenged the role of CD24 as BC stem cell marker and concluded that CD24 was not suitable to detect BC stem cell in vivo (Hacek et al 2021). We therefore rephrased the introduction to reflect this point and added new references (see lines 95 – 107 and side bars in the PDF “changes”).
- Variation of this biomarker is also noted in organoids. There are only three samples in the study which is not enough.
Ad 2: This critique is well taken and matches with point 3 of reviewer 1 (see above). Originally this study started with 7 BCO samples, 4 of which were high grads urothelial carcinoma-derived, 1 was low-grade carcinoma-derived, 1 was squamous cell carcinoma, and 1 was small cell carcinoma. But only the 3 BCO lines matched the inclusion criteria of long-term culture and providing sufficient materials for quantitative RT-PCR analyses and immuno-fluorescence. Analysis of the more frequent BCOs let’s say in the first or second passage would increase the numbers available for the short-term analysis but miss a key point of this study: expression of cell surface markers as function ot in vitro culture time. In the revision we have discussed this limitation of the study (compare lines 477 – 490). But we will of course continue our research along these lines.
Round 2
Reviewer 3 Report
I'm happy with these changes.